# Fit-for-Purpose Information for Offshore Wind Farming Applications—Part-II: Gap Analysis and Recommendations

**Johannes Schulz-Stellenfleth** [1,*] , **Anouk Blauw** [2] , **Lauri Laakso** [3,4] , **Baptiste Mourre** [5] , **Jun She** [6] and **Henning Wehde** [7]

1   Helmholtz-Zentrum Hereon, Max-Planck-Str. 1, 21502 Geesthacht, Germany
2   Deltares, 2600 MH Delft, The Netherlands; anouk.blauw@deltares.nl
3   Finnish Meteorological Institute, Erik Palménin aukio 1, 00560 Helsinki, Finland; lauri.laakso@fmi.fi
4   Atmospheric Chemistry Research Group, Chemical Resource Beneficiation, North-West University, Potchefstroom 2531, South Africa
5   Balearic Islands Coastal Observing and Forecasting System (SOCIB), 07121 Palma, Spain; bmourre@socib.es
6   Danish Meteorological Institute, Lyngbyvej 100, 2100 Copenhagen, Denmark; js@dmi.dk
7   Institute of Marine Research, Nordnes, NO-5817 Bergen, Norway; henning.wehde@hi.no
*   Correspondence: johannes.schulz-stellenfleth@hereon.de

**Abstract:** Offshore wind energy installations in coastal areas have grown massively over the last decade. This development comes with a large number of technological, environmental, economic, and scientific challenges, which need to be addressed to make the use of offshore wind energy sustainable. One important component in these optimization activities is suitable information from observations and numerical models. The purpose of this study is to analyze the gaps that exist in the present monitoring systems and their respective integration with models. This paper is the second part of two manuscripts and uses results from the first part about the requirements for different application fields. The present solutions to provide measurements for the required information products are described for several European countries with growing offshore wind operations. The gaps are then identified and discussed in different contexts, like technology evolution, trans-European monitoring and modeling initiatives, legal aspects, and cooperation between industry and science. The monitoring gaps are further quantified in terms of missing observed quantities, spatial coverage, accuracy, and continuity. Strategies to fill the gaps are discussed, and respective recommendations are provided. The study shows that there are significant information deficiencies that need to be addressed to ensure the economical and environmentally friendly growth of the offshore wind farm sector. It was also found that many of these gaps are related to insufficient information about connectivities, e.g., concerning the interactions of wind farms from different countries or the coupling between physical and biological processes.

**Keywords:** offshore renewable energies; fit-for-purpose information products; monitoring systems; data assimilation; observation system optimization

## 1. Introduction

The offshore wind energy sector has grown massively worldwide since the first wind park at sea was commissioned in Denmark in 1991. The building of offshore wind parks has accelerated over the last decade, and this development will likely continue at least until the middle of this century [1]. According to the European Union (EU) Strategy on Offshore Renewable Energy [2], the installed offshore wind capacity in Europe will grow by a factor of five, from 12 GW today to 60 GW by 2030. The Global Wind Energy Council (GWEC) Market Intelligence forecasts that by 2030, more than 205 GW of new offshore wind capacity will be added globally, including at least 6.2 GW of floating offshore wind power [3]. This development is driven by very ambitious and concrete goals defined by politics, e.g., to achieve climate neutrality by 2050 in Europe. In Germany, a target of 30 GW installed

offshore wind power by 2030 is written in law, which means an almost quadrupling of the capacity that existed in 2022. In the wider European context, the development of new offshore wind farm (OWF) activities also takes place in new areas with currently little or no existing OWFs. These areas include the Northern Baltic Sea and Mediterranean Sea, with challenges specific to these regions. As the development of new OWFs is expected to be fast and local legislation may be behind, best practices from other, previously developed regions should be utilized, and approaches and impacts potentially harmful to the society and environment should be avoided.

The growth of the OWF sector comes with a large number of scientific and technological challenges [4,5], e.g., in the fields of:

o　　OWF design and planning;
o　　Installation of OWFs;
o　　Operation and maintenance (O&M) of OWFs;
o　　Environmental impact assessments;
o　　Dismantling, repowering, or recycling of OWFs.

The sustainable evolution of offshore wind energy technology in terms of cost efficiency and environmental impacts requires detailed information about the two-way interaction between the OWFs and their environment [6]. A key component to meeting this demand is dedicated monitoring systems that are integrated with up-to-date numerical models for the environment and the technology. The combination of simulation tools and observations for specific-use cases has gained new attention in the context of digital twins [7], which are seen as an efficient tool for decision making. In [8], an overview of the requirements for integrated information was provided from observations and modeling. In the current study, we perform a gap analysis to evaluate to what extent the current observation and modeling capabilities are sufficient for providing the required information during different lifetime phases of OWFs. We identify what capabilities are still missing and how these can potentially be developed.

Gap analysis is applied in different fields, e.g., in the private sector, and is seen as a powerful tool to develop and grow business [9]. More specifically, it helps to:

● 　Define priorities;
● 　Identify areas for improvement;
● 　Allocate resources in a strategic way;
● 　Measure progress in an objective way;
● 　Achieve goals within a given time frame.

A variety of observation gap analysis methods have been investigated in the field of operational oceanography, which can be divided into two categories: one is to assess data adequacy for reconstructing a four-dimensional, continuous ocean state [10,11]; the other is to assess data adequacy to fit for certain given purposes, e.g., operational forecast, environmental assessment or offshore wind farm siting [12]. The first type of method quantitatively evaluates a data impact index, e.g., "effective coverage", "sampling error", or "initial uncertainty", for a given sampling scheme. Observing system simulation experiments (OSSEs) fall into this category as well. Here, the quality of observations is assessed in terms of the ability to improve model forecasts in a data assimilation scheme using general statistical parameters like RMSE or using a more basic approach based on assumptions about the correlation structure of the model errors [11]. The fit-for-purpose gap analysis, on the other hand, consists of three stages. The first stage is to define an application area, e.g., offshore wind farm siting and tailored products needed for this service; then, all the available observations and modeling approaches will be used to generate the tailored products; finally, adequacy of the observations is assessed according to experiences in generating the products. This method can be either qualitative or quantitative. A fit-for-purpose data adequacy assessment was performed for OWF siting in the Baltic Sea [13]. In EMODnet (European Marine Observation and Data Network) CheckPoint projects, data adequacy in multiple application areas, such as OWF siting, oil slick forecasting, river discharge, climate

change, and fishery management, was assessed for European regional seas [12]. However, these applications were analyzed separately.

In this study, we apply a fit-for-purpose gap analysis with reference to requirements for the OWF sector identified in [8]. In that study demands concerning observations were identified and discussed for the six application fields, which differ in characteristic temporal and spatial time scales. The focus of the study was on aspects with high connectivity either across spatial scales, system compartments (e.g., atmosphere/ocean), or ecosystems.

(1) Operation and maintenance (O&M);
(2) Submarine cables;
(3) Wake and lee effects;
(4) Transport and security;
(5) Contamination;
(6) Ecological impacts.

Gap analyses have been performed in the context of offshore wind energy in a number of studies. For example, [14] performed a study about monitoring gaps in the ecosystem in the Dogger Bank region. Data gaps with regard to offshore wind resource assessments and optimal designs were discussed in [15,16]. A gap analysis concerning rules, regulations, and standards is provided by [17,18]. Missing knowledge about the impacts of sea power cables on the environment is discussed in [19]. An early report about guidelines for data acquisition to support marine environmental assessments for offshore renewable energy projects was given by [20]. The general importance of the topic was discussed in various documents, e.g., a recent report by the European Marine Board [21] stated that the "lack of sustained funding for Ocean observations and marine monitoring has created the lack of baseline knowledge across European seas needed to develop the ORE (Offshore Renewable Energies) required by European ambitions."

The present study extends and complements the existing investigations in different ways, e.g.,

(1) Oceanic and air/sea interaction aspects are put into the focus;
(2) The discussion is centered around fit-for-purpose information products for different use cases;
(3) Particular focus is put on high connectivity aspects, which are of high importance for decisions about trans-European monitoring strategies;
(4) The discussion includes physical, chemical, and ecosystem aspects.

The manuscript is structured as follows: In Section 2, a short description is provided of the methodology applied for the gap analysis. In Section 3, a very brief introduction is presented of the six use cases and the existing modeling and monitoring capacities are summarized. Different European countries are used as examples to explain the present situation. In Section 4, gaps in the existing monitoring systems and model integrations are summarized. In Section 5, these gaps are discussed in a larger context and recommendations are formulated. Finally, Section 6 provides a summary and conclusions.

## 2. Methodology for Gap Analysis and Input Data

In this section, a brief introduction is given to the general concept of a gap analysis. This includes the objectives, as well as characteristic properties of the method as an optimization tool. In addition, several aspects are discussed, which have to be considered when using this approach in the context of observation systems in the offshore wind energy sector.

The gap analysis conducted in this study follows general principles used in different contexts and in particular in the business sector [22]. Four basic steps need to be considered in the analysis (see Figure 1):

1. A desirable target scenario has to be defined;
2. The current situation has to be assessed;
3. Gaps have to be identified;
4. Strategies to fill the gaps have to be developed.

The target scenario should comply with the SMART principle, i.e., it should be specific, measurable, achievable, relevant, and time-bound. The target scenarios for offshore wind energy are very specific for Europe, including definitions of very ambitious timelines. The growth of OWF installations can be measured in terms of installed power, but it is clear that this metric is not sufficient for a holistic assessment of the technology. Apart from the energy costs for the final consumer, the safety of the energy supply and potential societal and environmental impacts have to be considered as well. The definition of respective metrics to measure the fitness of the associated monitoring systems and progress in the implementations is even more challenging. The final goal should be to answer the following question:

- How well do the observations fit for the purposes of applications in terms of cost efficiency and environmental friendliness in technology and operations, and where are the gaps?

The most underdeveloped part of such assessments is the quantification and evaluation of environmental damages in relation to economic benefits. This is related to the definition of concepts like "green economy", which still requires further sharpening [23]. In this study, we will not enter into the broader political and ethical dimension of this debate but rather concentrate on the more technical aspects. We will, however, include discussions on data policies as well as the communication between different actors in the offshore wind sector because they are of direct relevance to the efficient use and evolution of monitoring systems.

The gap analysis presented here is based on the identification of requirements given in [8] and covers a variety of aspects:

- Availability and suitability of sensors;
- Observation coverage in time and space;
- Observation accuracies;
- Observation consistency (metadata, validation procedures, etc.);
- Use of observations in combination with models for model optimization, assimilation, and validation.

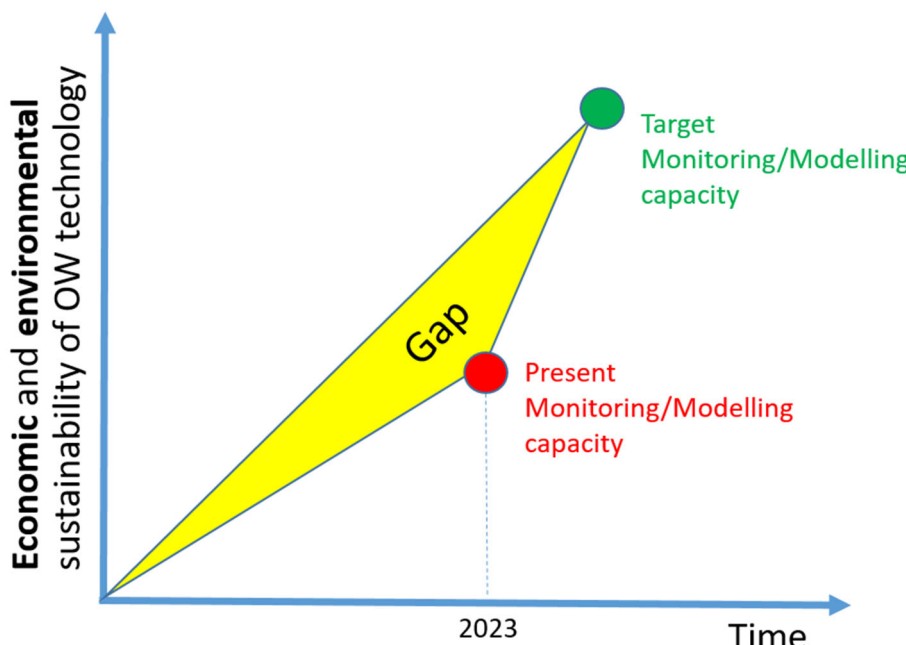

**Figure 1.** Diagram illustrating the major components of the gap analysis.

Information about the present status of available measurements was gathered from the existing literature as well as freely accessible information. Metadata of satellite observations in European seas are obtained from the Copernicus Marine Environment Monitoring

Service (CMEMS). For in situ observations, metadata are obtained from both EMODnet and national databases, which consist of in situ observations from operational agencies, environmental monitoring, geological survey, and fishery monitoring from Denmark, Finland, Germany, Netherland, Norway, and Spain. In addition, data from research infrastructures such as Danubius, ICOS-OTC, EURO-ARGO, and the suite of JERICO (Joint European Research Infrastructure of Coastal Observatories) projects are used, highlighting the essential importance of the ESFRI (European Strategy Forum on Research Infrastructures) activities for sustainable development of OWFs and use of ocean energies. For some application areas, research and commercial observations are also used. Information about OWF installation plans was gathered from different sources, e.g., OSPAR (Oslo and Paris Conventions) and documents issued by national agencies, e.g., the Federal Maritime and Hydrographic Agency in Germany (BSH) [24]. The authors are taking part in the JERICO-S3 (Joint European Research Infrastructure of Coastal Observatories: Science, Service, Sustainability) project and are familiar with the latest developments in the ocean monitoring sector both on the European level and on the national level.

The focus of the analysis is on gaps concerning information products, that require knowledge about processes with high connectivity. We are using the term connectivity in a wider sense, such that it includes both connectivity in the spatial dimension and connectivity across different processes including human activities.

## 3. Existing Monitoring and Modeling Capacity

In this section, a very short introduction is given to the different use cases, and the main overall information requirements identified in [8] are summarized. The main technological and environmental components, as well as a number of key parameters addressed in this study, are visualized in Figure 2. Subsequently, present solutions to provide the required observations in combination with model simulations are presented. The solutions are discussed using the situation in a number of European countries as an example. Experiences from existing solutions are analyzed, and recommendations for the further development of OWF in other regions are provided. The evolution of OWF is diverse in European seas, and the discussion uses a limited number of regions with interesting developments as examples, namely the southern North and Baltic Seas, the northern North/Norwegian and Baltic Sea, and the Mediterranean Sea.

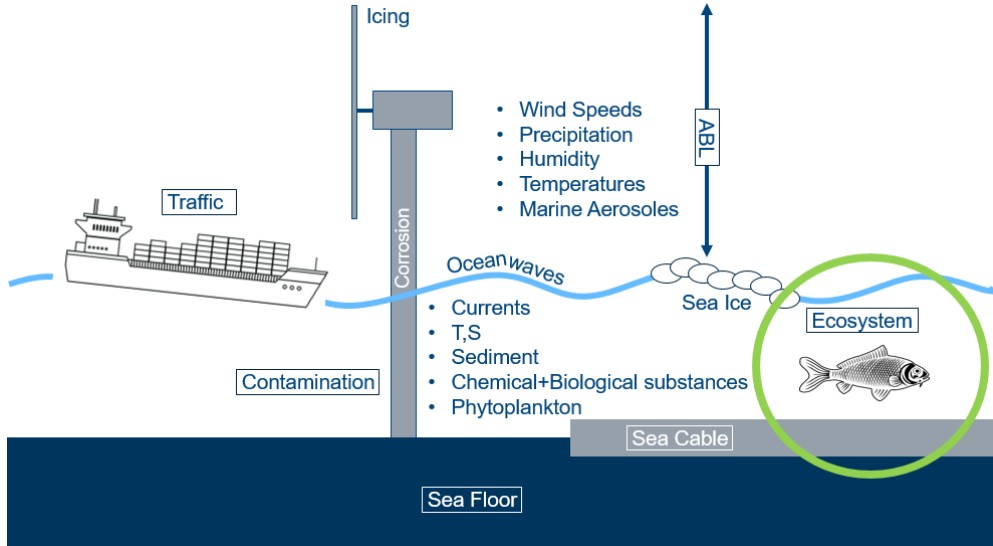

**Figure 2.** Main components of the technology and the environment considered in the gap analysis for the offshore wind sector.

*3.1. OWF Inspection and Maintenance*

Operation and maintenance (O&M) costs constitute a substantial part of the total financial investment required for OWF project lifecycles [25]. With about 14–30% of the total expenditure spent on O&M [26], the optimization of the respective procedures and technologies is of vital importance to make offshore wind profitable and economically sustainable. Ship operations are a particularly relevant component in this context since the costs for vessels sum up to about 50% of the total O&M costs, with typically about six visits required per year for mostly minor O&M activities at each turbine [27,28]. For the optimized use of ship time, reliable information about environmental conditions is crucial [29]. For example, depending on the ship type, limits exist for the significant wave height $H_s$, at which crew transfer vessels (CTVs) are allowed to transfer personnel to the turbines (e.g., 1 m for Monohull or 1.2 m for Catamaran). Environmental information is furthermore required in the context of predictive maintenance, which is of relevance in the context of corrosion [30] or structural health [31]. For the monitoring of the aging process, dedicated measurements of the structure response to wind, waves, or currents are of interest, e.g., obtained from strain sensors or accelerometers [32]. In particular, with regard to corrosion protection, possible environmental impacts are of concern as well. The general topic of pollution will be discussed in more detail in Section 3.5.

3.1.1. Existing Monitoring Solutions for O&M

The most important variables for O&M are waves and currents. In the European seas, CMEMS provides 30-year altimetry data (significant wave height, wind speed, and sea level anomaly). Regular along-track products have a resolution of about 7 km, while 5 Hz products provide 1.2 km resolution data, which greatly increases the availability of coastal, especially nearshore, observations. EMODnet has data from 219 wave buoys in the Baltic–North Sea. For currents, the Baltic–North Sea is well covered by 116 mooring stations. In addition, there are 11 HF radars in the North Sea [33]. This provides a base for solid model validation in the Baltic–North Sea scale. A more detailed overview of the observations in German waters is provided below.

In situ: The core element of the in situ observation system along the German coast is the network of tide gauges with about 19 stations in the German Bight and 32 stations in the Baltic. A significant number of additional tide gauges can be found upstream the rivers (e.g., Elbe, Weser, Ems). Nine stations of the MARNET network (MARitimes UmweltmessNETzwerk) operated by BSH measure salinity, temperature, and surface currents. Furthermore, about nine wave buoys provide sea state information [34]. Within the pre-operational Coastal Observing System for Northern and Arctic Seas (COSYNA), a number of stationary and mobile platforms measure physical, geochemical, biological, and key sediment variables [35]. The research center Hereon operates three HF radar stations to measure surface currents in the German Bight [36], and it has operated gliders for certain periods as well as FerryBox systems both on ships and as stationary systems. Regular measurement campaigns are performed with ships (e.g., Ludwig Prandtl), e.g., including scanfish measurements. Dedicated airborne campaigns to analyze the OWF impacts on sea state were conducted by the University of Braunschweig [37,38].

Very few open-access operational measurements are taken dedicated to the offshore windfarm topic. One exception is the FINO-1 platform located next to the first German offshore wind park Alpha Ventus. Because of the rapid growth of installations in the vicinity, this platform is not suitable any more to measure free stream conditions.

Remote sensing: In general, the operational use for OWF applications in coastal areas is still quite rare. For Germany most of the use is in the context of scientific studies or in test setups at operational centers. Hereon has used satellite SST and altimeter data for validation and assimilation of circulation and ocean wave models along the German coast. Optical satellite data were used to study sediment transport processes and for data assimilation. Furthermore, satellite radar data were used to study high-resolution wind fields around OWFs, e.g., wake effects. BSH is using satellite data (e.g., SST) in pre-operational setups for



data assimilation. Most of the satellite data are accessed via CMEMS (Copernicus Marine Environment Monitoring Service), but in some cases (e.g., TerraSAR-X or CFOSAT), other channels have to be used as well.

### 3.1.2. Existing Modeling Solutions for O&M

The operational model forecast for German coasts is performed by BSH for the circulation part. Operational ocean wave forecasts with 3 days lead time are provided by the German Weather Service (DWD). The core element of the BSH model system is the 1 km BSHcmod 3D circulation model for the German coastal water, which is two-way nested into a coarser North Sea/Baltic Sea model. DWD uses the WAM model in combination with the atmospheric ICON model. Six-day wave forecasts with about 1.5 km resolution are available for the North West Shelf area from the European Copernicus system [39]. Hereon is using various model setups for the coastal German waters with a strong emphasis on research aspects related to the coupling between atmosphere, wave, and ocean circulation. The standard models used in this context are NEMO, WAM, and the unstructured grid model SCHISM, which is suitable for analyzing small-scale processes in estuaries and rivers or around offshore wind farms [40]. Strong cooperation exists between Hereon, the University of Hamburg, and the Max-Planck Institute for Meteorology (MPI) in the context of multiscale ocean modeling, e.g., combining the MPI-OM and ICON models with SCHISM. Offshore wind farms were included in parameterized form in the atmospheric COSMO model as well as in the ocean circulation model SCHISM, which are both part of the Hereon GCOAST system [41]. OWFs are, however, not yet included in operational models.

### 3.2. Protection of Submarine Cables

For protection of submarine cables, the key information product is sediment layer thickness above the cable. In Part I [8], a cost-effective solution for generating this product was proposed, i.e., through integrated use of survey observations and coupled ocean–wave–sediment modeling tools. Below, we analyze the availability and assess the adequacy of community observations and modeling capacity for performing the survey-modeling integrated approach for predicting areas with high mobile sediments and burial depth changing rate. Here, the "community observation" means data measured by public agencies, or private or citizen data openly available.

In this application, Danish and adjacent waters are used as an example in the analysis. Danish EEZ is located in both Baltic and North Seas, and is part of Baltic–North Sea transition waters. In order to simulate sediment transport in the Danish EEZ, both the Baltic–North Sea area and Danish EEZ waters should be resolved, especially high resolution with 1 km or smaller grid is needed. Input data for a sediment model, including both bedload and suspended sediment, consist of bathymetry, currents near seabed, waves, sediment discharges from land, median grain size, bed slope, sediment density, salinity, and temperature. Among these variables, the most important information is waves, currents, sediment density and grain size, and bed slope. Sediment layer thickness itself is one of the model outputs, and observations are needed to validate the model.

### 3.2.1. Existing Monitoring for Submarine Cable Protection

For all applications, as long as an integrated monitoring-modeling approach is used, bathymetry and river discharges are the two basic input datasets.

Bathymetry: in European seas, EMODnet Bathymetry provides gridded bathymetry data with about 115 m resolution. In shallow water, more recent bathymetry was mapped using data from satellites, e.g., Sentinel 2. DHI-Group offers such data at DHI bathymetry portal as commercial products with spatial resolution of 10 m and 2 m, as well as an uncertainty measure that indicates accuracy for each data point. This method can produce bathymetry in different times using frequently revisited Sentinel 2 (since early 2016). For Danish waters, data with 50 m resolution or higher in 50 m can be obtained from

Danish Geodata Styrelsen. In addition, bathymetry can be measured using acoustic devices operated from ships [42].

Lateral sediment flux to the sea: this includes sediments from the rivers and coastal erosion. The data are needed as lateral forcing in the sediment models, including both suspended and bedload components. In [43], 79 major rivers were used for the Baltic–North Sea region. For the area of the North Sea, the main source of information were the Delft Hydraulics study Contaminant retention in North Sea estuaries [44] and the OSPAR (Oslo and Paris Conventions) report on Riverine Inputs and Direct Discharges (RID) [45]. This report is regularly updated. The most recent one is published in 2021 [46]. The RID database, which is maintained in the Norwegian Institute for Bioeconomy Research (NIBIO), can be accessed online. The riverine SPM inputs in Sweden were obtained from [47]. SPM inputs in major rivers from other Baltic Sea countries can be obtained from Global River Water Quality Archive (GRQA) [48].

Coastal erosion is another source of sediment entering the sea, for example, the English cliffs of Suffolk, Norfolk, and Holderness. The west coast of Denmark has soft cliffs consisting of very fine particles such as clays and fine sands. The shoreline in this region has been retreating at a speed of 0.5–4 m per year due to coastal erosion during the past 40 years [49]. Severe fine sand transport occurs mainly during storms. The sea level rise and increasing extreme events, e.g., flooding and storm surge, can increase risks of the nearshore section of the submarine cables and the cable stations [50].

Seabed substrate: in the Baltic–North Sea, EMODnet Geology provides a seabed substrate map in a scale of 1:100,000 with an EUNIS category. In Danish waters, GEUS provides a seabed substrate map in a scale of 1:250,000, with seven substrate categories. The sediment classification expresses the sediment type of the upper 0.50 m of the seabed. Each sediment class is defined based on the specific grain size distribution. In addition, information about sediment distributions can be obtained using acoustic instruments mounted on ships [51,52]

SPM concentration in the sea: hourly SPM concentration has been measured intensively using, among others, the SmartBuoy, FerryBox, and glider methodologies in the European seas since 2000 [53]. Turbidity data from many mooring buoys can also be transformed into SPM concentration. Some of those data are available, for example, in the REPHY database [54] for the North Atlantic Shelf Seas. In the Baltic Sea, there are several research datasets, containing a few hundred samples that cover the western, eastern, and northern Baltic Sea. In inner Danish waters, SPM concentration has been measured in three transections. These in situ observations are available from EMODnet Geology and can be used to validate the satellite and model products. For surface SPM concentration, CMEMS provides comprehensive satellite products in the Baltic–North Sea, including an open sea product in 4 km resolution, an offshore product of 300 m resolution (up to 200 km from the coast) and a nearshore product of 100 m resolution. However, the SPM products were only validated using in situ measurements from the REPHY (Observation and Monitoring Network for Phytoplankton and Hydrology in coastal waters) database.

Sedimentation rate, sediment layer thickness, critical shear stress: observations on seabed net sedimentation rate and/or sediment layer thickness are required for model calibration and validation. EMODnet Geology has collected such data, which well cover the Baltic Sea, the Skagerrak, and the Norwegian Trench. However, there are little data existing in the Danish EEZ (Exclusive Economic Zone) and the open North Sea. In addition, information on the critical shear stress for moving gravel sediments are rarely available. These data or information are only available from geological surveys for the industrial sector or individual research such as in [55].

Waves and currents: In general, current and wave measurements are rarely existing for European waters, although currents profiles near seabed are especially valuable for validating sediment transport models. However, such observations are mainly made by the oil and gas exploration industrial sector.

### 3.2.2. Existing Modeling Capacities for Submarine Cables

The models required for submarine cable protection are coupled ocean–wave–sediment transport models. Such models are already available, e.g., Coupled Ocean–Atmosphere–Wave–Sediment Transport Modeling System (COWAST) [56], developed by USGS (United States Geological Survey). The sediment transport model includes both cohesive and non-cohesive sediment dynamics [57]. Another model is a finite element coastal ocean–wave model SCHISM (Semi-implicit Cross-scale Hydroscience Integrated System Model)—WWM (Zhang et al., 2016a,b [58,59]). A sediment transport module is also included. The model system has been applied in the North Sea and the Baltic Sea [58,60]. However, these models have not been applied and validated for submarine cable protection.

In addition, knowledge of critical shear stress and settling velocity of sediments with different grain sizes are still of high uncertainty in the sediment transport module [61]. Observations are needed to improve the parameterizations.

### 3.3. Wake and Lee Effects

For applications related to the wake and lee effects of OWFs, the key information product is the impact of OWFs on winds, ocean conditions, waves, and sediment transport. In Part I [8], an integrated monitoring-modeling approach was proposed. The models required include weather, ocean, wave, and sediment transport models, which can resolve multiple scales ranging from individual OWF scale to coastal connectivity scale and multi-farm and cross-border scale. These models will not resolve individual turbines. Instead, the effects of individual turbines are parameterized according to the models' grid sizes. Such parameterizations can be developed using observations or combined with turbine-resolving very high-resolution computational fluid dynamics (CFD) simulations such as large eddy simulations (LES). Observations in the farm site and surrounding waters are required to derive the turbine-effect parameterizations and to calibrate and validate the CFD and ocean–wave–sediment transport models.

### 3.3.1. Existing Monitoring Solutions for Wake and Lee Effects

Observations to study and assess wake and lee effects are gathered by OWF operators, research programs, operational monitoring agencies, environmental monitoring, and coastal agencies. Sea state, sea level, currents, and winds within OWFs are often monitored exclusively by OWF operators. These data are confidential and can be used for research after a non-disclosure agreement (NDA) is signed. Nacelle wind speed and operational variables are measured by SCADA (Supervisory Control And Data Acquisition) systems from all turbines, together with wave data from buoys and winds from ground-based LIDAR data. Research projects may also obtain permission to carry out multi-disciplinary monitoring activities, including physical, wave, sediment, biogeochemical, and biological monitoring. For EC-funded projects, research data should be released as soon as possible, following the FAIR (find, access, interoperate, and reuse) principles. In particular, multi-disciplinary datasets obtained from research projects dedicated to studying the wake and lee effects will be very useful for deriving and validating parameterizations of turbine and OWF impacts. One example is the three FINO research platforms, which provide hourly meteorological and oceanographic observations in German EEZ waters, including wind profile data from a mast of 103 m high. Operational and coastal agencies are responsible for carrying out operational monitoring on, e.g., sea level, waves currents, and winds in the coastal waters, using fixed platforms of tide gauge stations, moorings and coastal morphological stations, and FerryBox. Sometimes, these stations are in the outskirts of OWFs, e.g., MARNET buoys operated by BSH, tide gauge, FerryBox, and HF radar networks in European coastal seas, so the observations can be used to quantify the wake and lee effects and validate the OWF impact-resolving models. Environmental monitoring is regular low-frequency (4–24 times a year) sampling in air, seawater, biota, and seabed. These data can be used for model validation. In Europe, operational and environmental monitoring observations and part of the research observations have been collected and

centrally disseminated by EMODnet. Furthermore, OWF lee effects were analyzed in the framework of dedicated airborne campaigns [62,63]. Airborne data were used as well to evaluate OWF parameterizations in atmospheric models [64].

Regular observations of atmospheric wakes are provided by satellite synthetic aperture radar (SAR) data as flown on the European Sentinel-1/2 platforms. Analysis of these data and derivation of empirical parameters to describe the spatial structure of wakes have been presented in a number of studies (e.g., [65,66]). The measurements have big potential because of the high spatial resolution (<100 m) and the large coverage (>100 km). The observations are however limited by relatively poor temporal sampling caused by the dusk/dawn acquisition cycle with overflights every couple of days. We are not aware of the use of these data on a routine basis for wake monitoring.

Another standard measurement technique to study wakes in the atmosphere [67] is based on long-range Doppler light detection and ranging (lidar). These ground-based measurements have a smaller spatial but higher temporal resolution than satellite SAR systems.

### 3.3.2. Existing Modeling Solutions for Wake and Lee Effects

The models for assessing the wake and lee effects can be divided into two categories according to their grid resolution: OWF-resolving models with a grid size larger than the turbine foundation but smaller than the OWF coverage and turbine-resolving model with a grid size smaller than the radius of a turbine foundation. The atmospheric wake effects have been parameterized (e.g., [64,66,68,69]), and implemented in mesoscale numerical weather prediction (NWP) models. To our knowledge, these parameterizations are not yet included in operational models for forecast production. Sensitivity experiments of the OWF-resolving HARMONIE in the Baltic–North Sea region can reproduce the wake effects in the atmosphere [70]. A study concerning atmospheric OWF wakes for the North Seas using the COSMO model in combination with the Fitch parameterization was presented in [71]. Engineering models with simpler parameterizations and less computational costs are used in industry (e.g., [72]).

The impacts of atmospheric wakes on hydrodynamics and waves have been recently studied by [73,74] using an unstructured grid model SCHISM and by [75] using COWAST coupled atmosphere–ocean–wave models. However, these models do not include turbine parameterization in hydrodynamic and wave models. An early analysis of the impacts of OWFs on sea state was provided in [76]. The study concluded that the strongest effects are associated with the reduced wind forcing. Additional impacts are related to reflection and diffraction of waves at the foundation structure as well as wave dissipation caused by friction at the piles. For the effects on hydrodynamics, a parameterization of the additional mixing and friction due to a turbine structure is developed as an extension of the k—$\varepsilon$ two-equation turbulence closure model [77]. A high-resolution Reynolds-averaged Navier–Stokes (RANS) model of the local scale is used to calibrate this parameterization. Unstructured grid ocean models have also been used in turbine-resolving impact modeling studies [78]. For OWF impacts on waves, the turbine can be treated as unresolved obstacles (UOST), and parameterization on UOST has already been included in popular wave models such as WAM, WWIII, and WWM [79].

### 3.4. Transport and Security

OWFs have impacts on observations and logistics on the sea. There are different types of impacts, which require more research and further observations. Here, we focus on two of them influencing the transport and safety sector in the Baltic Sea. First, while the research focus has mainly been on the impacts of sea ice on mechanical construction of OWFs (e.g., [80]), the large offshore installations influence the environment by changing the natural motion of ice fields. This has an impact on wintertime marine transport, as especially in the Northern Baltic Sea, conditions for winter maritime transport change. The changes in ice fields also influence marine ecosystem due to the impacts on mixing, sea–air exchange, and underwater light conditions. Secondly, marine surveillance is based

on a coastal radar network, which is strongly impacted by the large OWFs, as the OWFs create reflections and shadowing of objects [81]. This limits the construction of OWF, especially in the Gulf of Finland, but also in areas in the vicinity of Kaliningrad. The OWFs also influence the functioning of the weather radars and limit both wind and precipitation observations over the sea areas. The decreased accuracy of observations needed for weather forecasting increases the potential security risks related to lack of accurate environmental information. A comprehensive analysis of the situation in the Finnish territorial waters has been published (in Finnish, with abstract in English) by [82].

In accordance with the oil and gas industry, ship traffic inside the OWF is regulated and activities such as fishery is very limited. There are several research projects ongoing aiming to optimize the multiuse of marine space. The recent evolvement of the political situation in Europe makes it also necessary to take terroristic activity and the impact of militaristic actions into account including ship traffic inside OWFs (e.g., fishery, terrorism, military).

In the following, we concentrate on the Northern Baltic Sea, with a rapidly increasing number of OWFs in the near future and seasonal ice conditions, which causes additional challenges for the OWF sector.

### 3.4.1. Existing Monitoring Solutions for Transport and Security

Currently, there are practically no OWFs in the Northern Baltic Sea [8]. However, the number of planned OWFs is very large and the situation will change rapidly in only a few year time scale. The existing observing network is an optimized balance between the current needs and available financial resources. The current observing network consists of a limited number of marine weather stations, mainly manual ice observations, wave and temperature buoys, few FerryBox lines, and Argo (Array for Real-Time Geostrophic Oceanography) floats. These are supported by remote sensing methods utilizing X-band coastal radars, AIS (automatic identification system) network for ship tracking, weather radars, satellite remote sensing products, and irregular monitoring cruises. Some additional data are obtained through other observations, like maritime cameras and hydrophones, but it is typically not available for public research or forecasting purposes. Additionally, the current political situation impacts the reliability of AIS data as there are cases both with falsified AIS signals and dark vessels (i.e., AIS transponders turned off). All these security aspects combined also influence the protection of seabed cables (Section 3.2), whether damaged accidentally (environmental conditions) or intentionally (hostile human activities). Thus, several overlapping and independent methods are needed [83].

### 3.4.2. Existing Modeling Solutions for Transport and Security

The ocean models used in the Northern Baltic Sea are developed for a range of societal needs on transport, security, and environments. These models include an operative hydrodynamic model with sea ice forecasting capabilities (NEMO-LIM3) and wave models (WAM, SWAN). The atmospheric modeling, including wind fields, is carried out with the Harmonie–Arome NWP model. These models produce sea state and weather fields necessary for environmental analysis and transport sector forecasts. They are also used as modeling input values for assessing the impacts of OWFs on (radar) electromagnetic signal propagation over the sea.

Sea ice forecast is an important product for transport and security related to the OWF industry. In the Baltic Sea, several sea ice models such as LIM, CICE, HELMI, and HBM-ICE have been developed and coupled with hydrodynamic models to provide an operational forecast of the sea ice. Assimilation of sea ice concentration observations is now available in the CMEMS BAL MFC forecasting system [84]. In Finland and Sweden, the model forecast and sea ice charting are combined for providing the ice service for operations in the sea.

In case of collisions of ships or damage of submarine cables, there might be severe leaks of oil, gas, or chemicals from the vessels. Three-dimensional drift modeling of pollutants will be needed. The on-demand oil drift models have been operational in most of the Baltic Sea countries [85]. The similar drift models have also been used for search and rescue.

Oil drift model in pack sea ice has been developed by [86], which is very useful for the Northern Baltic Sea.

### 3.5. Contamination

There are two main aspects to be considered to evaluate the impact of contaminants associated with OWF installations. On the one hand, metal and chemical concentrations need to be monitored and modeled in the vicinity of the emitting sources. On the other hand, ocean currents need to be observed and/or simulated to assess the regional dispersion of these contaminants toward the rest of the oceanic basin.

Corrosion protection systems used for OWF turbines might be responsible for the release of aluminum, cadmium, zinc, and indium into the ocean) [87–91], with potential toxic effects on marine life. Aluminum, cadmium, and indium are non-essential metals for marine organisms. When introduced artificially in an environment with a relatively high concentration, aluminum can negatively affect important regulation and respiratory functions of adult fishes [92]. Cadmium is recognized as an environmentally highly toxic metal that can accumulate in marine flora and fauna, be transmitted through the food web and eventually affect human bodies [93,94]. While zinc is a necessary element for the functioning of marine organisms, it also represents a risk of toxicity with increased concentration [95]. Once in the ocean, these metals were found to be able to latch onto floating plastics, favoring ingestion by marine organisms and insertion into the food web, thus representing a threat to ecosystems at large [96,97]. Organic compounds with high toxicity, including biosphenol A [98], are also part of the substances associated with corrosion protection measures that may end up in the ocean due to material damage or weathering processes [91]. While the effect of these emissions from corrosion protection systems is probably relatively low compared to other sources such as rivers, atmospheric depositions, or fossil fuel industries [88], the potential toxic risk for marine organisms makes it necessary to monitor the presence of these different components in the vicinity of the wind farms.

Ocean currents then have the capacity to transport these contaminants over large distances. While they may have a dispersive effect that progressively reduces their concentrations as long as they are transported over the basin, currents may also accumulate them in specific locations due to oceanographic or topographic singularities. Knowing the possible trajectories of these contaminants once released at the OWF sites is crucial to characterizing the oceanic connectivity, evaluating the impact of these installations over entire ocean basins, and understanding the path of these substances across administrative boundaries.

### 3.5.1. Existing Monitoring Solutions for Contamination

Monitoring the concentration of metals and other contaminants typically requires taking samples of either water, bottom sediments, or tissues of marine organisms. The concentration of metals dissolved in seawater can be measured by collecting water samples and analyzing them after filtering in the laboratory. Since the toxicity may also depend on water hardness, pH, dissolved organic carbon, and temperature conditions, these complementary chemical parameters should also be monitored. The analysis of samples of sediments and tissues can provide additional information on the presence of metals on the ocean floor, and the potential impact of bioaccumulation processes in marine organisms. While sample analysis techniques are available, they remain quite costly and, to the best of our knowledge, they have not been implemented for operational automated measurements.

Concerning ocean currents, a routine monitoring of large-scale features is performed by satellite altimeters through the measurement of sea surface height anomalies and subsequent determination of associated geostrophic currents. However, these observations suffer from limitations in the coastal zone and only represent spatial scales larger than a few tens of kilometers. In coastal areas, high-frequency radars (HFRs), installed on the shore, have the capacity to measure the surface flows with a kilometer-scale resolution and

a spatial coverage of a few tens of kilometers from the coast [99,100]. When covering wind farms areas, HFR measurements represent an ideal solution to monitor ocean currents and water pathways in the vicinity of the OWF. Surface drifters may also be deployed in the area of interest to infer drifting trajectories from the OWF infrastructure, but they might not necessarily provide robust information since their trajectory strongly depends on the ocean conditions at the time of the deployment given the high spatio-temporal variability of ocean currents in the coastal zones.

### 3.5.2. Existing Modeling Solutions for Contamination

Hydrodynamic modeling provides a tool to represent the evolution of ocean currents over wide areas and characterize the ocean connectivity at the regional scale. Nowadays, simulations and predictions of ocean currents are generated operationally with a spatial resolution close to 1 km in some regions of the world (e.g., https://marine.copernicus.eu/, accessed on 1 June 2023) [101–104]. The incorporation in the models of the information provided by routine and multi-platform observations (from satellite, profiling floats, underwater gliders, HFR) through data assimilation provides a way to constrain the simulations to be as close as possible to the observed conditions. Hydrodynamic-wave coupling can also be implemented to enlarge the range of resolved processes and in particular represent the wave-induced drift at the ocean surface. Telescopic model nesting then also allows for refining the spatial resolution in limited areas of specific interest. Sediment transport modules are also useful to model the sedimentation and resuspension of particles. On top of this, Lagrangian modeling [105] can be applied to calculate trajectories from simulated currents and explore the spatio-temporal ocean connectivity at the regional scale.

### 3.6. Ecological Impacts of OWFs

The development of OWFs has impacts on the marine ecological environment [89,106]. There are aspects that, in general, can be associated with positive impacts, such as that renewable energy helps reduce greenhouse gas emissions and mitigate the climate change effect. In addition, OWF development can contribute to the development of artificial reefs that provide opportunities for benthic organisms to develop a higher diversity than in unchanged environments. Those areas are also potentially attracting several fish species, leading to new environments for development and increased biodiversity (i.e., [107]). Those positive effects are potentially in exchange with the negative effects caused by the development of OWFs, where noise and vibration under construction and in the drift phase can potentially impact marine species such as fish, mammals, and invertebrates [108,109]. In addition, OWFs can pose a collision risk for birds and bats, especially during migration or when placed in important feeding or breeding areas (i.e., [110]). Furthermore, the installation of wind turbines and the associated infrastructure (e.g., cables and substations) can cause physical habitat alteration and loss. For example, the installation of OWF can disrupt the seabed and benthic ecosystems and have an impact on the behavior of the marine species via the change in the electromagnetic fields, which is caused by undersea cables transmitting electricity from OWFs [111,112]. Furthermore, the currents around monopiles in OWFs increase turbulent mixing, potentially leading to the break-up of stratification [113,114], increased turbidity [115], and changes in primary production [6,116,117].

### 3.6.1. Existing Monitoring Solutions for Ecological Impacts

Monitoring the ecological impacts of offshore wind farms is crucial to assessing and mitigating potential effects on marine ecosystems and for model validation. Underwater acoustic monitoring systems are used to assess the impact of noise generated during OWF construction and drift on marine organisms following procedures developed and implemented (i.e., [118]). These systems can track and analyze sound levels, underwater noise propagation, and the behavior of marine species in response to noise. Visual and radar systems are employed to monitor bird and bat activity around wind farms. These systems can detect and track the flight paths of birds and bats to assess collision risks.

Additionally, bird and bat observers may be stationed on vessels or offshore platforms to conduct real-time monitoring. Video monitoring using underwater cameras and remotely operated vehicles (ROVs) allows for direct observation of the marine environment around OWF. These surveys can assess the presence, behavior, and interactions of marine species, including fish, marine mammals, and benthic organisms. Satellite imagery and remote sensing techniques can provide valuable information on changes in sea surface temperature, total suspended matter and phytoplankton biomass, and the distribution of marine species at the water surface over larger spatial scales. Observations of vertical profiles of physical and water quality variables require profiling buoys or profiles observed from ships. Benthic surveys are conducted to sample and monitor the seabed and associated organisms in the vicinity of OWF (for example, [51]). The conduction of eDNA analysis involves collecting and analyzing water samples to detect and identify genetic material shed by organisms in the environment. It can provide information on the presence, abundance, and diversity of species and potentially provide information on changes in behavior.

3.6.2. Existing Modeling Solutions for Ecological Impacts

The construction and operation of OWFs can have significant impacts on marine ecosystems and the habitats of marine organisms. Those impacts are mainly caused by changes in (1) noise, (2) habitat, (3) electromagnetic fields, and (4) water quality. Spatially explicit frameworks to analyze the integrated effects of wind farms on the marine environment aiming to evaluate how wind farms can contribute to the protection of the marine environment through strategic and economically viable location choices are developed and applied for quite a long time (i.e., [119]). Systematic methods for mapping how increased pressures from human activities may cause cumulative ecological effects on marine ecosystems are developed [120]. Those frameworks aim to provide answers regarding the integrated effect. A couple of such frameworks are established for specific regions. In the Netherlands, the Deltares model D-FLOW-FM-DCSM is used to evaluate the potential effects of future OWFs on currents, vertical mixing, suspended sediment concentrations, phytoplankton dynamics, and benthic filter-feeders [121]. For validation of this model, vertical profiles of temperature, salinity, suspended matter, and phytoplankton are the main gaps in required observations for model validation. In the area of water quality modeling, there is a long tradition of developing models for the human impact on the marine ecosystem (i.e., [116,122]).

Those integrated frameworks depend crucially on the realistic modeling of the specific impact factors on the ecosystem. In order to be able to calculate the sound level at a given distance from the source, it is important to have sufficient knowledge of the parameters that must be included in such a model [123], such as sound source, water depth, bottom topography, properties of the bottom and water column (density, sound speed, and attenuation). Parameters that are often not sufficiently known. For habitat changes, there exists a variety of model approaches for the specific components of the ecosystem (i.e., fishes: [124]). The modeling of the electromagnetic field and changes via the implementation of OWFs is in a premature state, and many approaches are taken from terrestrial applications. However, [125] have conducted modeling evaluations investigating EMF (electromagnetic fields) by subsea power cables. For all types of ecological models, system understanding of the long-term impacts of OWFs is the main gap for further model development and testing. We are only starting to observe and understand these impacts as the implementation of OWFs is under development.

## 4. Gap Analysis

In the following, gaps are identified for all six use cases. The analysis is structured along different gap categories, e.g., gaps in accessibility and availability of observed variables, as well as deficiencies in spatial and temporal sampling or in model-observation integration. It is obvious that this analysis can never be totally objective. It is, however, a view that is shared among the authors, who come from six European countries and

who were involved in various projects with industry and agency involvement. We are also aware that the assessment of gaps will change over time because of the extremely dynamic situation in terms of technology developments and the largely unpredictable political boundary conditions. As the authors are not representing the entire offshore wind sector, we are not trying to make strong statements regarding priorities, but we rather see this analysis as a contribution to a broader discussion among industry, agencies, politics, and research that is necessary on a European level and beyond.

A condensed overview of gaps regarding monitoring and modeling is provided in Tables 1 and 2, respectively. We will discuss these deficiencies in more detail in the following.

**Table 1.** Monitoring gaps in the OWF sector for different use cases.

| Variable | Use Case | Gaps |
|---|---|---|
| Bathymetry | O&M | More regular surveys desirable to optimize wave forecasts |
| | Protection of sea cables | Detailed bathymetry near cables (for accurate bed slope calculation) not accessible |
| | Wake and lee effects | Detailed OWF bathymetry is still challenging to obtain, but this is not the main source of modeling errors |
| | Transport and security | n.a. |
| | Contamination | n.a. |
| | Ecological impacts | Limited data availability on stability of sediments as habitat for benthic organisms |
| Shoreline | O&M | No major gaps |
| | Protection of sea cables | No major gaps |
| | Wake and lee effects | Regularly updated shorelines; more observations desirable in Wadden Sea areas because of impacts on ABL |
| | Transport and security | n.a. |
| | Contamination | n.a. |
| | Ecological impacts | n.a. |
| Wave height | O&M | More consistent wave observations on coastal and regional scale desirable, including accuracy information |
| | Protection of sea cables | Dedicated wave observations near cables are needed |
| | Wake and lee effects | Dedicated wave observations in the wakes |
| | Transport and security | Availability will improve radar performance estimates and sea state forecasting close to OWFs |
| | Contamination | n.a. |
| | Ecological impacts | No major gaps |
| 2D wave spectra | O&M | Homogeneous spatial distribution of 2D observations, including OWF sites desirable |
| | Protection of sea cables | Dedicated wave observations near cables are needed |
| | Wake and lee effects | Dedicated wave observations in the wakes |
| | Transport and security | Availability will improve radar performance estimates and sea state forecasting close to OWFs |
| | Contamination | n.a. |
| | Ecological impacts | n.a. |

**Table 1.** *Cont.*

| Variable | Use Case | Gaps |
|---|---|---|
| Surface winds | O&M | To improve coupled wave and atmosphere models, more wind profile observations are required inside and outside OWFs |
| | Protection of sea cables | No major gaps |
| | Wake and lee effects | Observations in the wakes |
| | Transport and security | OWFs weather radar shadowing effects need to be compensated with additional observations |
| | Contamination | n.a. |
| | Ecological impacts | n.a. |
| Wind profiles | O&M | To improve coupled wave and atmosphere models, more wind profile observations are required inside and outside OWFs |
| | Protection of sea cables | No major gaps |
| | Wake and lee effects | Observations inside OWFs and in surrounding areas |
| | Transport and security | Changes in vertical wind profiles and turbulence may influence radar signal propagation close to the sea surface<br>OWFs weather radar shadowing effects need to be compensated with additional observations |
| | Contamination | n.a. |
| | Ecological impacts | n.a. |
| Atmospheric boundary layer parameters (including icing and humidity) | O&M | More vertical profiles of temperature and humidity are needed to improve ABL stability and icing conditions in forecast models, as well as corrosion prediction |
| | Protection of sea cables | No major gaps |
| | Wake and lee effects | Observations inside OWFs |
| | Transport and security | Vertical temperature and humidity profile observations necessary for modeling electromagnetic signal propagation |
| | Contamination | n.a. |
| | Ecological impacts | n.a. |
| Precipitation | O&M | Standardized measurements suitable for training of ML models insufficient |
| | Protection of sea cables | n.a. |
| | Wake and lee effects | n.a. |
| | Transport and security | OWFs weather radar shadowing effects need to be compensated with additional observations |
| | Contamination | n.a. |
| | Ecological impacts | n.a. |

**Table 1.** *Cont.*

| Variable | Use Case | Gaps |
|---|---|---|
| Surface current | O&M | More observation required in particular in the vicinity of OWFs |
| | Protection of sea cables | Nearshore currents in brackish waters |
| | Wake and lee effects | Incomplete coverage of nearshore currents (esp. in brackish waters) |
| | Transport and security | Additional observations in and around OWF |
| | Contamination | Incomplete coverage of coastal areas by HF radars |
| | Ecological impacts | n.a. |
| Current profiles | O&M | It is debatable whether more profile information is needed to better capture abrasion processes |
| | Protection of sea cables | Currents near seabed in cable areas |
| | Wake and lee effects | Currents and turbulence measurements in the wakes and nearby OWFs |
| | Transport and security | n.a. |
| | Contamination | n.a. |
| | Ecological impacts | Limited data availability for both inside and outside of OWF for comparison |
| T&S | O&M | More observations required, in particular, near OWFs for corrosion prediction |
| | Protection of sea cables | No major gaps |
| | Wake and lee effects | Inside and nearby OWFs, especially in wakes |
| | Transport and security | Inside and nearby OWFs |
| | Contamination | Local observations required to (1) constrain simulations of hydrodynamics, and (2) evaluate toxicity of contaminants |
| | Ecological impacts | Limited data availability of vertical profile data and long time series for trend detection |
| Underwater sound/noise | O&M | n.a. |
| | Protection of sea cables | n.a. |
| | Wake and lee effects | n.a. |
| | Transport and security | Additional underwater noise observations may be needed |
| | Contamination | n.a. |
| | Ecological impacts | Limited data availabillity |
| Land-based sediment discharge | O&M | n.a. |
| | Protection of sea cables | Lack of daily or monthly data |
| | Wake and lee effects | Lack of daily observations |
| | Transport and security | n.a. |
| | Contamination | n.a. |
| | Ecological impacts | n.a. |

| Variable | Use Case | Gaps |
|---|---|---|
| SPM concentrations and composition, settling velocity | O&M | n.a. |
| | Protection of sea cables | No major gaps |
| | Wake and lee effects | Need dedicated in situ data in wakes and lee area |
| | Transport and security | Changes in underwater visibility may impact the use of optical underwater methods |
| | Contamination | n.a. |
| | Ecological impacts | Limited data availability of vertical profile data and long time series |
| Seabed sediment properties (type, sedimentation, and erosion rate) | O&M | n.a. |
| | Protection of sea cables | Lack of regularly updated basin-scale dataset, esp. in cable areas |
| | Wake and lee effects | Need regularly updated data in OWFs and wake/lee impact areas |
| | Transport and security | Changes in seabed may need additional surveys |
| | Contamination | n.a. |
| | Ecological impacts | Limited availability of long time series to detect changes |
| Sea ice | O&M | More reliable observations needed in vicinity of OWFs |
| | Protection of sea cables | Lack of in situ ice thickness and fast ice data |
| | Wake and lee effects | Lack of in situ ice thickness and fast ice data |
| | Transport and security | More reliable observations needed in vicinity of OWFs |
| | Contamination | n.a. |
| | Ecological impacts | Limited data available for ice conditions impacting ecosystem |
| Concentration of Al, Zn, Cd, In, BBA, etc. | O&M | n.a. |
| | Protection of sea cables | n.a. |
| | Wake and lee effects | n.a. |
| | Transport and security | n.a. |
| | Contamination | Lack of regular measurements in the vicinity of OWF sites |
| | Ecological impacts | Lack of regular measurements in the vicinity of OWF sites |
| Concentrations of dissolved oxygen, pH, pCO2, alkalinity | O&M | More observations required for corrosion prediction |
| | Protection of sea cables | n.a. |
| | Wake and lee effects | n.a. |
| | Transport and security | n.a. |
| | Contamination | Lack of regular measurements in the vicinity of OWF sites |
| | Ecological impacts | Lack of long consistent time series for trend detection and interpretation |

**Table 1.** *Cont.*

| Variable | Use Case | Gaps |
|---|---|---|
| Plankton | O&M | n.a. |
| | Protection of sea cables | n.a. |
| | Wake and lee effects | n.a. |
| | Transport and security | n.a. |
| | Contamination | n.a. |
| | Ecological impacts | Limited availability of long consistent time series of primary production and species composition for trend detection and interpretation |
| Fish, marine mammals, birds | O&M | n.a. |
| | Protection of sea cables | n.a. |
| | Wake and lee effects | n.a. |
| | Transport and security | n.a. |
| | Contamination | n.a. |
| | Ecological impacts | Limited availability of long-term time series to assess changes in distribution around OWFs |

**Table 2.** Modeling gaps in the OWF sector for different use cases.

| Model | Use Case | Gaps |
|---|---|---|
| Hydrodynamic model | O&M | Atmospheric wakes not included in meteo forcing of operational ocean models |
| | Protection of sea cables | On-demand (re-locatable) modeling capacity is needed |
| | Wake and lee effects | Wake effects not included in operational weather and ocean models |
| | Transport and security | Accurate, combined hydrodynamic models needed for estimating impact of sea surface properties on radar signal propagation |
| | Contamination | High-resolution (<1 km) regional models constrained by observations |
| | Ecological impacts | Smooth coupling between high-resolution models in OWFs with larger-scale models |
| Wave model | O&M | Two-way coupled wave/atmosphere models with wake parameterization still not consolidated. Atmospheric wakes not included in operational forecast models |
| | Protection of sea cables | Wave-induced vertical momentum flux needs to be validated |
| | Wake and lee effects | Two-way coupled wave–atmosphere models with wake parameterization still not consolidated. Atmospheric wakes not included in operational forecast models |
| | Transport and security | Accurate information on wave properties inside OWF's needed for estimating sea clutter |
| | Contamination | High-resolution (<1 km) regional models in areas where they are not yet available |
| | Ecological impacts | No major gaps |

**Table 2.** *Cont.*

| Model | Use Case | Gaps |
|---|---|---|
| Weather model | O&M | Effects of OWFs on observations used in operational data assimilation schemes not considered so far |
| | Protection of sea cables | No major gaps |
| | Wake and lee effects | Wake effect-resolving operational forecast model is needed |
| | Transport and security | Accurate NWP modeling inside and in vicinity of OWFs needed for radar performance modeling |
| | Contamination | Wake effect-resolving operational forecast models would bring added value |
| | Ecological impacts | No major gaps |
| Metal pollutant modeling | O&M | Contamination models related to corrosion protection not mature (see also contamination use case) |
| | Protection of sea cables | n.a. |
| | Wake and lee effects | n.a. |
| | Transport and security | On-demand modeling capabilities in case of accidents not mature |
| | Contamination | Metal emission models from OWF infrastructures |
| | Ecological impacts | Metal emission models from OWF infrastructures |
| Suspend particulate matter model | O&M | n.a. |
| | Protection of sea cables | Need more validation and calibration for storm cases in shallow waters |
| | Wake and lee effects | Need more validation and calibration for storm cases in shallow waters |
| | Transport and security | n.a. |
| | Contamination | n.a. |
| | Ecological impacts | Validation of OWF impact on vertical profiles of SPM needed |
| Chemical pollutant modeling | O&M | Contamination models related to corrosion protection not mature (See also contamination use case) |
| | Protection of sea cables | n.a. |
| | Wake and lee effects | n.a. |
| | Transport and security | On-demand modeling capabilities in case of accidents not mature |
| | Contamination | Chemical emission models from WOF infrastructures |
| | Ecological impacts | Validation is needed |
| Seabed sediment model | O&M | n.a. |
| | Protection of sea cables | Estimate of critical shear stress needs further improvements |
| | Wake and lee effects | More validation and calibration needed in nearshore waters and storm cases |
| | Transport and security | n.a. |
| | Contamination | n.a. |
| | Ecological impacts | Interaction between biota and physical processes needs to be better understood |

**Table 2.** *Cont.*

| Model | Use Case | Gaps |
|---|---|---|
| BGC low trophic model | O&M | n.a. |
| | Protection of sea cables | n.a. |
| | Wake and lee effects | n.a. |
| | Transport and security | n.a. |
| | Contamination | n.a. |
| | Ecological impacts | Further validation is needed and coupling between OWF scale and ecosystem scale |
| Habitat model | O&M | n.a. |
| | Protection of sea cables | n.a. |
| | Wake and lee effects | n.a. |
| | Transport and security | See ecosystem use case |
| | Contamination | n.a. |
| | Ecological impacts | Further development and validation needed |
| High trophic food web model | O&M | n.a. |
| | Protection of sea cables | n.a. |
| | Wake and lee effects | n.a. |
| | Transport and security | See ecosystem use case |
| | Contamination | n.a. |
| | Ecological impacts | Processes yet insufficiently understood to be realistically modeled |

### 4.1. Gaps in the Accessibility of Observed Variables

In this section, two types of gaps are addressed. Firstly, gaps in data availability are discussed, which refer to relevant variables that are currently not observed at all. Secondly, the problem of data accessibility is analyzed, which refers to the obstacles encountered when trying to access existing datasets.

The effects of OWFs on the atmosphere and the ocean are strongly conditioned by processes in the atmospheric boundary layer (ABL). For example, the ABL stability has a big impact on the length of atmospheric wakes [65]. Currently there are very few measurements suitable to assess the state of the atmosphere (e.g., profiles of temperature, wind, and humidity). Furthermore, many of the available measurements, e.g., from FINO-1 are affected by the surrounding wind parks, i.e., they do not provide information on free stream conditions. For a better understanding and model representation of OWF interaction with the ocean it is paramount to have more information about fluxes of momentum and heat in the vicinity of the wind farms.

For the O&M use case, measurements of momentum and heat fluxes near the sea surface would contribute to optimizations of coupled atmosphere/wave/circulation models, which are required to provide reliable short-term forecasts of the conditions during maintenance operations. Furthermore, there is a lack of reliable measurements needed for predictive maintenance related to corrosion, in particular dissolved oxygen, sulfate, and pH. Dissolved oxygen and pH are provided by the CMEMS modeling system, but sulfate is not. Of particular concern with respect to corrosion are the pile segments, which are periodically wetting and drying due to wave impacts, as well as the structure above, which is affected by marine aerosols. For both processes, more detailed information on wave breaking and respective statistics in the vicinity of the wind parks is required.

For contamination assessment, concentrations of most of the contaminations (e.g., Al, Zn, BBA) in the OWF and surrounding areas have not been monitored. The data are needed in water samples and in benthic and pelagic bio-samples.

Accessibility to existing data related to offshore wind farm applications can be quite different according to which type of observations are concerned: operational, environmental, commercial, or research data. For operational data access, CMEMS, INS, TAC, and EMODnet have collected most of the in situ observations in Europe and made them freely available to the public. For observations from environmental monitoring, the data are also freely available via EMODnet, ICES (for the Baltic–North Sea), SeaDataNet, and national ocean data centers. However, these data are mainly sampled by research vessels and distributed in a delayed mode. The locations are not chosen to detect changes in environmental conditions due to OWFs. Some countries, e.g., Norway, Sweden, and Estonia, have initiated near real-time ship data, especially CTD (conductivity, temperature, and depth) data delivery. Other countries, such as Germany, Denmark, and Finland, have their CTD data available in a few weeks, while data from EMODnet chemistry, ICES, and SeaDataNet can only be available months to a couple of years after the monitoring. For OWF operational applications, e.g., O&M, near real-time access to the data is required, but this is mainly for metocean variables with high-frequency observations.

Commercial and research monitoring provides more data on a local scale compared to operational and environmental monitoring, i.e., within OWFs and surrounding areas. However, these data are more limited for public access. The data usages are often subjected to signing NDAs (non-disclosure agreements). In recent years, there have been some efforts for collecting and disseminating commercial and research observations related to OWFs. 4C Offshore (https://www.4coffshore.com/, accessed on 1 June 2023) provides worldwide offshore wind farm information with a membership fee. The Crown Estate Marine Data Exchange (MDE, https://www.marinedataexchange.co.uk/, accessed on 1 June 2023) holds data from a variety of industries, including marine aggregates, subsea cables, tidal and wave energy, offshore wind, and also data from research and evidence projects, which have grown to almost 300 TB of survey data from over 50 offshore projects across the U.K.; over 2600 survey campaigns covering over 15 survey themes, from geophysical data to marine mammal surveys.

According to the Crown Estate Data policy, regarding environmental data, despite the contractual position with regard to confidentiality, in general the Crown Estate will not release data relating to a particular project, until consent is awarded and the period for judicial review has passed. Once a firm consent decision has been determined, the data are effectively in the public domain, so generally will be released thereafter.

For physical survey data including geophysical and geotechnical data, the Crown Estate will hold survey data relating to geophysical, geotechnical, metocean, and meteorological data, confidentially until a Financial Investment Decision (FID), subject to a biannual review from the date of consent, where the time period between consent and FID is extended.

However, not all countries have an organized data collection and release system for OWF survey data as the Crown Estate in the U.K. Considering that the OWF applications need survey data and environmental data in OWFs, such data collection and dissemination mechanism is crucial. OWFs also measure metocean data, e.g., winds and waves in the farm in near real time. These data are usually held by the OWFs. They may be used for research purposes if an NDA is signed. For research at the ecosystem scale, observations from different countries need to be combined. A centralized EC focal point for OWFs to upload their publishable data or metadata should be available to facilitate OWF data sharing and exchange.

Offshore meteo-masts have been built up and operated to measure meteorological and oceanographic observations in the past decade for research purposes. These data have been well managed at the national level and made available for research. In Germany, three masts (FINO 1, 2, and 3) have been maintained since 2007, and data access can be made

at https://login.bsh.de/fachverfahren/, accessed on 1 June 2023 after registration. In the Netherlands, wind@sea (https://www.windopzee.net/en/wind-op-zee/, accessed on 1 June 2023) collects, processes, and makes data available at eight offshore wind farm sites. In Denmark, DTU has maintained a website http://www.winddata.com for collecting and disseminating wind data, including 75 datasets at present, mostly from Danish waters. However, there is no centralized focal point from EC to collect and disseminate research data in OWFs, especially from EU FP7, Horizon 2020 (H2020), and Horizon Europe (HEU) projects. For H2020 and HEU programs, projects are mandatory to deliver a Data Management Plan (DMP), which, in principle, ensures that a project-oriented data policy based on FAIR principles is in place. Efforts are needed for centralized data delivery and publication of these projects.

*4.2. Gaps in Spatial Data Sampling*

For the six applications analyzed in this study, observations are required at four different spatial scales: (S1) within OWFs, (S2) between OWFs and the coast, (S3) cross-OWFs, and (S4) across national borders.

The atmosphere and ocean dynamics around OWFs is characterized by a strong coupling of these different spatial scales. For example, the presence of the adjacent land has an impact on the land/sea wind speed gradients [126,127]. The length of atmospheric wakes can extend up to 100 km downstream and the impacts on waves can reach even farther. The inhomogeneous sampling of atmospheric and oceanic parameters existing at the moment is not able to provide a complete picture of the 3D dynamics around OWF.

With regard to the O&M use case, the very heterogeneous sampling of wave and atmospheric boundary layer information is not optimal. As pointed out before, ocean wave dynamics encompasses a large spectrum of spatial scales with high connectivity, and in order to optimize wave forecasts, e.g., using data assimilation, a more regular sampling would be highly beneficial. This situation will get even more challenging with growing OWF installations, which can potentially impact waves on all scales (S1–S4).

For seabed cable protection, data are mainly needed in S2–S4 scales, with a focus on sections along seabed cables. Observations are mainly managed by energy agencies. All sediment conditions along the cable lines are monitored regularly. However, this monitoring can be optimized. With validated models, one can predict the sediment layer thickness above the cable, identify the areas with high risk, and optimize the sampling strategy. This may reduce the cost of monitoring largely. For validating models, a suitable research database on bathymetry, currents, waves, sediment types and concentrations, sedimentation, and erosion rates in the cable area is needed. In the sediment survey along the cable lines, if possible, the integrated measurements for these variables should also be made.

For assessing and predicting wake and lee effects, observations of wind, currents, turbulence in the sea, and ABL, waves, and sediment concentration are needed in all four scales, especially in the wakes. At the current stage, the main priority is to fill the knowledge gaps on the impacts and develop high-quality weather–ocean–wave–sediment models, which can predict the wake and lee effects. Currently, there is a lack of dedicated observations in the S1 scale in wake areas. Existing data have a limited number of stations in a farm and often close to the turbine. This is not suitable for wake study. There is also a lack of profiles of water temperature, salinity, and currents in the wakes. Danish and German waters can be a suitable testbed for the wake and lee effect study, as there is an operational monitoring network combined with HF radar, ADCP (Acoustic Doppler Profiler), moorings, tide gauge stations, and FerryBox, which can complement the commercial and research datasets.

For transport and security applications in icing waters, operational observations, especially waves and sea ice (concentration, edge, type, drift, and thickness) data, are needed. It is still not clear how the turbines may affect ice formation and drifting, considering enhanced turbulence in the wakes. The interaction between ice and waves is also an

important process for correctly predicting the sea ice and waves. To fill the knowledge gaps, dedicated in situ measurements of sea ice and waves in offshore wind farms are required for model calibration and definition. The in situ sea ice thickness is also important to quantify and reduce the uncertainties of the satellite observations. Currently, in situ sea ice and wave measurements in icing waters in the Northern Baltic Sea are quite sparse.

For the assessment of ecological impacts, the main gaps in spatial data availability are observations of vertical profiles of physical and biochemical variables and consistent data over the whole gradient where ecological impacts can occur. This covers at least the wind farm itself, including the monopiles and the area in between (S1), as well as the wake, which often exceeds national borders (S2–S4).

For most of the considered use cases, there is a lack of simultaneous observations inside (S1) and outside (S2–S4) of offshore wind parks, e.g.:

- To assess and forecast the conditions for O&M-based observations in free stream conditions outside the areas influenced by wind farms;
- To assess sea surface properties (waves, SST, ice) marine boundary layer parameters, which are relevant for radar signal propagation in the transport and security context;
- Sea ice observations required for model validation and data assimilation are missing;
- Some observations, e.g., from weather and military radars, are compromised by offshore wind farms, and this needs to be compensated by other observations;
- To assess the ecological impacts of no-fishing zones in the wind farm areas;
- To assess local and regional environmental impacts of anti-corrosion measures, e.g., sacrificial anodes [91];
- To assess wake effects inside of OWFs as well as larger-scale effects associated with neighboring OWFs, including those in neighboring countries (S4).

For some of the application areas, there is also a deficit concerning the simultaneous observation of coastal gradients and observations inside of the wind farms, e.g.,:

- To relate potential chemical contamination by anti-corrosion measures to contamination by rivers;
- To improve the understanding of the interaction between coastal wind speed gradients and atmospheric wakes.

For larger-scale effects, e.g., long atmospheric wakes, transports of contaminants, or the connectivity of habitats, harmonized datasets across European countries (S4) are still lacking. As discussed in the previous section, this is related to ongoing challenges concerning regulations and interactions between industry, agencies, and research.

### 4.3. Gaps in Temporal Availability

There are three major time scales of relevance for the discussed use case: T1 (operational time scale of a few days), T2 (installation lifetimes of about 25 years), and T3 (climate change time scales of 30 years and beyond).

Ideally, OWF impact studies should make use of observations taken before the installations were built, in the operation phase, as well as after decommissioning or repowering (T2–T3). The reality is that for most OWF sites, consistent observations of this kind do not exist. It is recommended to start observations three years prior to the start of the OWF installation. What is urgently needed is to define respective monitoring strategies for the OWF installations, which are planned for the future. In addition, there is no consistent strategy for long-term monitoring, e.g., to track the ecological impacts and impacts of climate change on offshore wind energy (T3).

In O&M application, we focus on two activities: platform dismantling assessment and operational maintenance. The former needs mainly long-term (T3), high-frequency wave data in S1, while the latter needs near real-time (T1) metocean data, especially winds and waves, mainly in S1 but also S2–S4 areas. Currently, most of the European OWFs have their own wind and wave conditions monitored operationally with an update frequency of 10 min or 1 h in the S1 scale. For areas of S2–S4, since no maintenance operations

will be carried out, a combination of operational monitoring (in situ and satellite) and model prediction can meet most of the requirements. Furthermore, for the O&M use case the availability of near real-time data with short latency is critical for the optimization of short-term forecasts. Such access does, in fact, exist for many observations, e.g., from the MARNET stations operated by BSH. The lack of consistent information about observation accuracy is still an ongoing issue, however.

For ecological impact assessment applications, long-term biogeochemical, habitat, and biodiversity data are required (T2–T3). The sampling needs to be made before and after the installation of the OWFs. The focus should be first put on the OWF and surrounding area in order to enable OWF siting in an area causing minimal impact, then in the area of S2–S4. Existing observations for this purpose are made during surveys for impact assessment of OWFs and research projects. Long-term, sustainable observations for ecological impact assessment are still lacking.

In general, one can say that a strategy is missing to develop a balance between long-term, consistent measurements and more flexible monitoring activities, which can become necessary to look at unexpected environmental impacts, improve process understanding, or validate on-demand modeling systems.

Another challenge that still exists is a mismatch between spatial and temporal sampling. An extreme example is satellite radar measurements, which provide very high spatial resolution and coverage, but the temporal sampling is not sufficient to capture the dynamics of the observed processes. On the other hand, observations from fixed platforms provide sufficient temporal sampling, but the coverage is often so poor that processes like advection related to spatial gradients are not resolved at all.

### 4.4. Gaps in Observation/Model Integration

When an integrated modeling-monitoring approach is applied for information provision, the basic idea is that the monitoring should provide quality-assured observations to fit for the purpose of improving model quality while models, on the other hand, can be used to optimize the sampling strategy and improve the cost efficiency of the monitoring activities. The applications in this study can be divided into four categories: (i) operational service (O&M, transport and security), (ii) regular or long-term assessment (cable protection, contaminants, ecological impacts), (iii) applications with significant knowledge gaps (e.g., cable protection, wake and lee effects), and (iv) on-demand and what-if service (e.g., O&M, cable protection, transport and security, contamination). The requirements and gaps can be quite different among the four categories. For the operational services, a major concern is the timeliness and quality of the forecasts for, e.g., winds, waves, and currents. Major gaps for this category of applications are:

1. Observations used for data assimilation in operational forecast systems start to get affected by OWFs, and this is not yet taken into account in the modeling systems;
2. There is a lack of strategy about the use of observations taken by the wind farm operators, e.g., in data assimilation schemes;
3. There is a lack of suitable observations for model validation and parameter tuning;
4. Machine learning (ML) and artificial intelligence (AI) tools should be developed to improve the local forecast by integrating local OWF observations and forecasts; long-term local observations are therefore valuable for training and optimizing the algorithms.

For long-term assessments, regular and long-term information products are needed; thus, model-observation integration should serve this purpose. Major gaps in this area are:

1. There is a lack of strategy concerning long- and short-term measurements, e.g., required for improved process understanding and respective model representation, model parameter optimization, or operational data assimilation;
2. There is a lack of information about realistic pan-European future OWF installation scenarios that can be used for optimization of monitoring systems using OSSE approaches, as well as model scenario calculations.

For applications with knowledge gaps, the model-observation integration should serve the purpose of adding new knowledge, including calibrating and optimizing relevant model parameterizations. Major gaps in this area are:

1.  There is a lack of observations in the targeted areas, such as along cable lines or in wake and lee areas, which are needed for optimizing OWF parameterizations in the models;

2.  To understand processes such as sediment erosion in the seabed and wake and lee effects, integrated observations are needed. Targeted sampling strategies should be designed to fill the knowledge gaps and improve model parameterizations.

For the applications that need on-demand and/or what-if scenario service, e.g., in case of collision, search and rescue, and pollution, on-demand modeling tools and observations are required. Major gaps in this area are:

1.  Existing on-demand modeling systems, e.g., oil spill, search and rescue systems, should be dedicated to the OWF industry and, therefore, be able to integrate local observations;

2.  The integrated model-observation system should be developed to supply extra information based on simulations of what-if scenarios when a critical environmental condition is likely to be reached and a decision on the operations has to be made.

In addition, it is essential to have information on observation accuracies, which is particularly critical for applications in the O&M context, where decisions with large financial implications have to be taken based on monitoring and modeling information. In the wake and lee effect studies, since the mean impacts of OWFs on the winds and waves are just a few percent, accurate data on winds and waves are thus very important to calibrate and validate the models. For wind power forecasts, the required accuracy for wind speed information is 3% due to the cubic dependence of wind power on wind speed [128]). Currently, there is a significant gap both in the availability as well as the standardization of such information. Activities to improve this situation do exist (e.g., [34]) and should be extended significantly.

Concerning the O&M use case, the integration of observations and numerical models is still not well developed. One of the challenges is the very short time scale of wave dynamics and the domination of sea state errors by inaccuracies in the driving wind fields. This means that the observed errors have to be traced back in order to realize efficient data assimilation schemes. Furthermore, some of the errors are caused by the forcing of the model at the open boundaries, and respective corrections are not trivial. Observations near these boundaries would add much value to the data assimilation schemes. It appears that the combination of classical data assimilation schemes and machine learning (ML) approaches or pure ML techniques [129] has the potential to address these problems, and more research is required in this field. For the training of ML methods, quality control and consistency of large observation datasets become even more important.

## 5. Discussion and Recommendations

In the previous section, gaps were identified in observation systems as well as in the integration of measurements with numerical models. The analysis was structured along different OWF use cases and along different observation characteristics, e.g., spatial sampling.

It seems obvious that the evolution of the existing monitoring systems was driven by a number of use cases, which had high priority in the past. For example, tide gauges were necessary for the development of storm surge forecast systems. Likewise, wave buoys have been important components in coastal management system, e.g., in the context of coastal erosion, for a long time. More sophisticated measurements, e.g., acquired by ADCPs, have become necessary to validate 3D circulation models, which are key elements in drift forecasts.

There is a general trend in the modeling community toward stronger coupling of different physical, biological, and chemical model compartments, which is necessary

to capture interaction processes of practical importance, e.g., the Stokes contribution of waves to the currents. As explained before, coupled models are an absolute necessity to capture processes in the vicinity of offshore wind farms and to provide respective information products. These coupled modeling systems have increased complexity in terms of dynamics and numerical implementation, i.e., model validation has become an even more challenging task, with broader requirements concerning observation systems. In particular, the validation of fluxes (e.g., energy, momentum, substances) between different model compartments is of growing importance for the new generation of coupled modeling systems. Many of the observation gaps identified in the previous section are related to missing information about the connectivity of different processes in the ocean and the atmosphere. This is a major bottleneck for the further optimizations of modeling systems and fit-for-purpose information products.

Due to the increased computational capacity available today, there is also a trend to finer spatial model resolutions. Unstructured grid models, which allow grid cells of only a few meters in size near the coast, have almost become a standard. As the typical spacing between offshore wind turbines is about 1km or below, it is obvious that model simulations for the offshore wind sector require fine spatial grids to resolve interactions between offshore wind farms and the environment. The validation of high-resolution models leads to new challenges for observation systems as well. Either one has to make sure that the sensor matches the resolution of the model, or one has to apply appropriate statistical methods for the validation. A careful characterization of the measurement process, e.g., spatial and temporal integration windows, is of increasing importance to make models and observations comparable. Likewise, reliable information about systematic and stochastic observation errors is essential for the assessment and optimization of models.

The formulation of recommendations for the evolution of monitoring systems in the context of offshore wind farming is complicated by the fact that various actors in this sector have to be considered. In addition, there is a larger spectrum of instruments that are on the table to drive certain developments. In the following, we will focus on the following pathways:

- Optimization of regulations and policies concerning data acquisitions and sharing, obligatory data sharing;
- Incentives for monitoring technology developments;
- Identification of synergies with other user groups of observation data;
- Additional observations and modeling to fill the observing gaps due to OWF radar shadowing effects and changes in sea surface properties;
- Implementation of a dynamic trans-European platform for information exchange and identification of changing requirements;
- Platform for communication between industry, agencies, and researchers;
- Complementary research, in particular concerning model/observation integration toward the development of a digital twin for the two-way coupled system of technology and environment.

With regard to data sharing, regulations should be put in place that make sure that offshore wind farm operators do not have a competitive disadvantage by opening access to their observations. We think that a combination of three strategies should be applied:

- Regulations should be implemented to make sure that standard observations are made public by all wind farm operators. Starting with the opening of the historical datasets would already be a step in the right direction. The approach used in the U.K. can be used as a first guideline;
- It should be more transparent how the different actors in the offshore wind sector can benefit from data sharing. In this context, research should better quantify the potential improvements in forecasts on different spatial and temporal scales;
- Regulations should be adjusted to allow cross-border measurement campaigns, e.g., with aircraft, ocean gliders, AUV, or drones. These systems can often operate autonomously, which leads to additional regulation requirements.

It is important to note that some OWF operators have already started to publish their observations for non-commercial use, e.g., Ørsted, and this development should be further encouraged.

With regard to incentives for technological developments, we see a number of areas with much potential:

- Drone technologies, whether in the air (AAV), at the sea surface (ASV), or underwater (AUV), are seen as very flexible tools both in the technological (e.g., blade inspections) and also in the earth system context (e.g., measurements in the atmospheric boundary layer/sea surface/underwater). In particular, developments toward a full automatization of this technology could lead to a step change with regard to monitoring in the offshore wind sector. Apart from the technological challenges, this will also require adjustments on the regulative side and in legislation;
- There are many promising applications of machine learning techniques in the offshore wind sector, e.g., in the context of corrosion modeling. These approaches rely on big, consistent, and quality-controlled observation datasets. There should be more joint efforts of industry and research to produce such datasets with open access.

Concerning synergies with other user communities, we see much added value in the following strategies:

- The offshore wind community should team up with the operational weather and forecast community. Operational observations are already affected by OWFs, and these have to be included in parameterized form in operational models. This, in particular, requires information about the operational status of OWFs;
- It becomes increasingly important to assess cumulative environmental impacts originating from different technologies. We, therefore, see many benefits in the design of combined monitoring strategies, including sectors like shipping, fishing, oil and gas, and industry discharging into rivers. There are also obvious synergies with military monitoring programs that could be exploited more;
- Offshore wind farm sites are areas with a relatively high density of observations and are therefore interesting candidates as test and validation sites for satellite systems. This would also provide the opportunity to optimize satellite observing systems for offshore wind applications.

For the security and transport use case, the following issues should be addressed:

- The potential need for additional weather radars to compensate for shadowing effects, joint planning with neighboring countries;
- Additional surveillance radars in OWFs shadow-specific areas;
- Additional sea ice observations (thickness, forces) are needed, especially in the beginning, to study the impacts of OWFs on sea ice;
- Additional vertical wind measurements; precipitation on marine weather stations; new marine weather stations;
- Wind turbines act as wind sensors: the data can be used to fill the measurement gaps due to the shadowing effects;
- Implementation of OWF-module to HARMONIE METCOOP NWP [69,130] and other regional high-resolution NWP models;
- Small-scale ice model development;
- Improved OWF parameterizations for radar signal propagation models.

With regard to a trans-European information platform, it is recommended that:

- The platform should contain consistent and updated information about the status and concrete future plans concerning OWF installations in Europe. This information should be sufficient to allow the integration of these installations into operational models and model scenario calculations;
- The platform should contain information in the form of datasets or interactive information systems, which allow wind farm operators and agencies to learn from the experiences, e.g., concerning environmental impacts in other regions;

- The platform should contain observation datasets, which are suitable for studying environmental conditions before and after offshore wind parks were built;
- The platform should define and contain observation datasets, which are suitable for long-term analysis of climate change impacts on the offshore wind sector;
- The platform should provide information about best practices for quality control of observation data and the definition as well as estimation of observation accuracies;
- The platform should provide best practice information on optimized OWF siting (i.e., siting with a minimal environmental impact and best coexistence with other industries).

With regard to a platform for communication between industry, agencies, and researchers, we see much potential in the following strategies:

- There should be a platform with continuity for the communication between industry, agencies, and researchers, which goes beyond the typical three-year cycle of national and European research projects. We think that this will help to build trust between these groups, and it will contribute to longer-term strategic planning, e.g., of scientific measurement activities;
- There has to be a continuous update and exchange of information about industry requirements, legislative frameworks, and new research developments.

With regard to complementary research and modeling activities, we have the following recommendations:

- The approach of on-demand modeling is seen as a very efficient tool to react quickly and in a flexible way to emerging new challenges, e.g., unexpected environmental impacts. This requires a respective modeling infrastructure and model interface harmonization;
- More dedicated observations should be gathered to optimize and validate coupled modeling systems, which are essential to capture the two-way interaction between the installations and the environment. Particular deficits exist in the atmospheric boundary layer and for ecosystems;
- OWFs have to be included in operational weather and ocean forecast models. Neglecting these installations will not only disregard the environmental effects of the OWFs, but also compromise the use of operational observations, which are impacted by the turbines. Data from OWFs would also help in filling the observing gaps due to radar shadowing effects;
- Integration of cross-border modeling and observation systems should be implemented to study and assess the impacts of OWFs on neighboring countries. This is also important to develop respective legislative frameworks related to, e.g., environmental impacts and ecosystems;
- The development of OWFs is currently going much faster than the development of observations, modeling, and understanding of their ecological impacts. This bears the risk that we only understand their ecological impacts after it is too late to reduce the number of OWFs in our coastal waters. By sharing data and experiences from existing OWFs, we can speed up the development of understanding, which would allow some time for adaptive management.

As a final comment one should say that the "static" view of a classical gap analysis as depicted in Figure 1 is to some extent oversimplifying the situation in the offshore wind energy sector. This is because of (1) the lack of process understanding, (2) fast technological developments, and (3) unpredictable dynamics in economic market developments and politics. This means that the design of monitoring systems for this sector should have considerable flexibility to allow for later adjustments concerning commercial focus areas and research priorities.

## 6. Summary and Conclusions

A gap analysis was presented for observation systems and respective integrations with numerical models in the context of fit-for-purpose information products required in the

offshore wind energy sector. The study is the second part of two papers, with the first one concentrating on the identification of requirements for six use cases. It was explained that gap analysis is a powerful tool to optimize decision processes by enforcing the development of clear ideas about target scenarios and the transparent assessment of the initial situation. The study also discussed the challenges of applying this tool in the context of offshore wind energy. One key challenge is the balancing of economic and environmental target definitions because this includes discussions about values and ethical aspects that require a broader discussion in society, i.e., this is not a purely scientific issue.

The study provided an overview of the monitoring and modeling solutions that are presently used to provide information products for the offshore wind community. It became quite clear that the observation and model systems used today have evolved due to requirements associated with a number of standard applications, e.g., storm surge forecasts or wave predictions for shipping. It also appeared that the monitoring of ecosystem parameters is less mature than respective systems for the measurement of physical quantities.

By comparing the present situation with the requirements identified in [8], gaps were identified, which were structured along different categories, e.g., spatial and temporal sampling or data availability and accessibility. Many of the identified gaps have to do with the fact that the existing monitoring systems are not adequate to capture characteristic length scales of today's offshore wind farms, e.g., related to the spacing of turbines. This means that different types of wake effects and turbine impacts on the environment cannot be assessed appropriately with the available observations. In addition, OWFs create new types of physical, chemical, and biological processes, which are not captured by the present monitoring systems at all, e.g., the generation of turbulence by turbine structures in the water and the atmosphere. Furthermore, it was discussed that most of the fit-for-purpose information products for the offshore energy sector have to include various types of connectivity aspects, e.g., the continuum of land, wind farm, and open ocean spatial scales. Likewise, the treatment of most optimization problems occurring in offshore windfarming requires detailed knowledge about interaction processes between different earth system compartments, e.g., the atmosphere, the ocean, the sea floor, and the ecosystem. There is still a lack of suitable measurements for this purpose, although information about these coupling mechanisms is also highly relevant in other contexts, e.g., climate change. There are also still observations missing to identify, understand, and predict two-way interactions between the technology and the environment. This has become increasingly challenging because of the rapid development of OWF installations in terms of turbine size and OWF coverage. It was also found that with regard to temporal sampling, a measurement strategy is missing to assess the environmental conditions before and after windfarms were installed. The issue is of growing urgency since locations not impacted by OWFs are increasingly hard to find.

Finally, a number of recommendations to fill the gaps were formulated. These include different technological aspects, e.g., autonomous systems like drones, but also suggestions concerning data policies and cooperations between science and industry. Due to the large-scale interactions of OWFs with the environment and also among each other, the development of measurement strategies across country borders was identified as an essential step forward. It is foreseeable that this step will also be of vital importance for a further synchronization and optimization of the energy system on a larger scale, e.g., across Europe. Another important recommendation concerns the exploitation of synergies by identifying common interests and requirements in different communities and sectors, e.g., the OWF community and operational forecast centers.

Finally, it is important to emphasize that this study is meant as a contribution to a discussion, which needs to be continued and extended. The task at hand is challenging not only because of the complexity and the rapid evolution of technology but also because of the diversity of the different communities that have to be brought together to find suitable solutions for the future. The experience in the past has shown that the respective

communication and synchronization processes take time and that makes a structured and transparent approach even more important.

**Author Contributions:** Conceptualization: J.S.-S. and J.S.; methodology: J.S. and J.S.-S.; analysis: J.S.-S. was responsible for operation and maintenance related continents, J.S. was responsible for submarine cable protection and wake and lee effects, L.L. was responsible for maritime safety in icing waters and radar aspects, B.M. was responsible for contamination, A.B. and H.W. were responsible for OWF impacts on habitat, NIS, fish, sea birds, and marine mammals; writing—original draft preparation, all; writing—review and editing, all. All authors have read and agreed to the published version of the manuscript.

**Funding:** The presented work was financed by the European Union in the framework of the project JERICO-S3 (Joint European Research Infrastructure of Coastal Observatories: Science, Service, Sustainability) (grant agreement ID: 871153). The work of L.L. has also been supported by the Academy of Finland, project number 338150, "Enabling forecasts on radar performance in marine environment".

**Institutional Review Board Statement:** Not applicable.

**Informed Consent Statement:** Not applicable.

**Data Availability Statement:** No new data were created or analyzed in this study. Data sharing is not applicable to this article.

**Conflicts of Interest:** The authors declare no conflict of interest.

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
