# Peer review of "Fit-for-Purpose Information for Offshore Wind Farming Applications—Part-II: Gap Analysis and Recommendations"

_jmse, doi:10.3390/jmse11091817_

Round 1
Reviewer 1 Report
The authors have presented a paper in the Gap Analysis, and a series of recommendations for monitoring and modelling offshore wind farming applications is provided. The paper deals with an interesting issue. There are some aspects of the paper which must be improved. With the aim of helping the authors, I provide some comments below about the encountered problems and possible solutions:
Abstract and keywords:
Do not include citations or mentions to other publications or authors in the abstract.
The introduction of the analysed problem in the abstract is too long. Please reduce provided information to just one or two sentences.
Avoid using, as a keyword, terms already used in the title. Deleted keywords included in the title and provided new keywords.
Introduction:
In the initial paragraphs of the introduction, there is a lack of references. Please provide more references to justify the affirmation in the introduction. Having at least 3 references per paragraph can be a good approximation.
The introduction is extremely long. Consider adding subsections.
Figure 1 should be moved. In the current position, it is dividing a paragraph.
Methodology for Gap Analysis and input data:
At the begging of the section, please add a short paragraph summarising the content of the section.
Paragraph lines 186 to 189 need some references.
Existing monitoring and modelling capacity:
I would be recommendable to add a scheme of the analysed parameters in this section in order to facilitate the comprehension of the content of this section.
For the following subsections, updated references are needed. Particularly in the " Seabed substrate", "SPM concentration", and "Waves and currents", the authors have to provide existing sensors for monitoring these variables.
The authors have to consider an alternative format for the content presented in Tables 1 and 2. Currently, they are consuming a lot of space, and their format does not facilitate their reading. The problem is particularly relevant in Table 1.
Discussion:
The discussion section must be modified. Currently, it is only a compendium of bullet points summarising the ideas. A discussion must be composed of a series of well-structured paragraphs providing ideas and justifying them with references or solid data. Thus, I suggest changing the title of the section to General Findings.
A conclusion section is needed in which the authors summarise the main conclusions of their paper and detail future work.
Reviewer 2 Report
The paper gives a exhaustive gap analysis of the observation and modelling capabilities to provide the necessary information during the different phases of the lifetime of Offshore Wind Farms. The information, references and recommendations make the paper a valuable tool for institutions, researches and industry in order to organizing data sharing, incentivize new developments in data acquisition technologies, facilitate synergies with other communities, improve the safety of maritime transport, foment the creation of trans European information platforms, promote platforms for communication between industries, agencies and researchers and encourage complementary research and modelling activities.
Only minor spelling errors and missing references have been detected:
Misspellings.
Page 27, line 998. Four categories have been presented.
Page 29, line 1088. “We think that a combination of two strategies should be applied”, this phrase is followed by three strategies.
Page 30, line 1038. The Security and Transport use case is not among the pathways indicated in lines 1066-1080 in page 29.
Page 31, line 1191. Please correct “actvities"
Page 32, line 1210. Please correct “boarder”
References
The references in the text (lines 239 and 240) to Dalgic et al. (2014a and b) are Dalgic et al. (2015a and b) in de References.
Page 16, line 708. Please complete the date of the reference Hutchison et al. (2021).
The reference Reese et al. (2022) is not indicated in the references.
The reference van Hellemont and Ruddick, (2014) is not indicated in the references.
The reference Minuzi and Leandro, (2023) is not indicated in the references.
The reference HASKONINGDHV NEDERLAND B.V. do not appear in the text.
Ther reference Håkanson, L., Mikrenska, M., Petrov, K., Foster, I.D.L., 2005, do not appear in the text.
The reference Lund-Hansen LC & C Christiansen (2008) do not appear in the text.
The reference Räty, A., Itäisen Suomen … do not appear in the text.
The reference Timco, G.W and Burden, R.P., do not appear in the text.
The reference Trockel D, Rodriguez-Alegre I, Barrick D, Whelan C, do not appear in the text.

Quality of English language is ok
Round 2
Reviewer 1 Report
The authors have addressed my comments.